# The EmojiGrid as a rating tool for the affective appraisal of touch

**Alexander Toet**[1]*, **Jan B. F. van Erp**[1,2]

**1** Perceptual and Cognitive Systems, TNO, Soesterberg, the Netherlands, **2** Research Group Human Media Interaction, University of Twente, Enschede, the Netherlands

☯ These authors contributed equally to this work.
* lex.toet@tno.nl

**Data Availability Statement:** All relevant data are within the paper and its Supporting Information files.

**Funding:** The authors received no specific funding for this work.

## Abstract

In this study we evaluate the convergent validity of a new graphical self-report tool (the EmojiGrid) for the affective appraisal of perceived touch events. The EmojiGrid is a square grid labeled with facial icons (emoji) showing different levels of valence and arousal. The EmojiGrid is language independent and efficient (a single click suffices to report both valence and arousal), making it a practical instrument for studies on affective appraisal. We previously showed that participants can intuitively and reliably report their affective appraisal (valence and arousal) of visual, auditory and olfactory stimuli using the EmojiGrid, even without additional (verbal) instructions. However, because touch events can be bidirectional and dynamic, these previous results cannot be generalized to the touch domain. In this study, participants reported their affective appraisal of video clips showing different interpersonal (social) and object-based touch events, using either the validated 9-point SAM (Self-Assessment Mannikin) scale or the EmojiGrid. The valence ratings obtained with the EmojiGrid and the SAM are in excellent agreement. The arousal ratings show good agreement for object-based touch and moderate agreement for social touch. For social touch and at more extreme levels of valence, the EmojiGrid appears more sensitive to arousal than the SAM. We conclude that the EmojiGrid can also serve as a valid and efficient graphical self-report instrument to measure human affective response to a wide range of tactile signals.

## 1 Introduction

Next to serving us to discriminate material and object properties, our sense of touch also has hedonic and arousing qualities [1–4]. For instance, light, soft stroking and soft and smooth materials (e.g., fabrics) are typically perceived as pleasant and soothing, while heavy, hard stroking and stiff, rough, or coarse materials are experienced as unpleasant and arousing [2, 5]. Affective touch has been defined as tactile processing with a hedonic or emotional component [6]. Affective touch plays a significant role in social communication [7]. Interpersonal or social touch [8] has a strong emotional valence that can either be positive (when expressing support, reassurance, affection or attraction [9]) or negative (conveying anger, frustration, disappointment [10]). Affective touch can significantly affect social interactions [11]. For example, a soft

**Competing interests:** The authors have declared that no competing interests exist.

touch on the forearm can lead to more favorable evaluations of the toucher [12], a light touch on the back can help to persuade people [13], and caressing touch (e.g., holding hands, hugging, kissing, cuddling, and massaging) can influence our physical and emotional well-being [14].

Recent studies have shown that discriminative and affective touch are processed by orthogonal somatosensory subsystems [15, 16]) that are already established early in life [17, 18]. The importance of touch in social communication is further highlighted by the fact that the human skin has receptors that appear specifically tuned to process some varieties of (caressing) affective social touch ("*the skin as a social organ*" [7]) in addition to those for discriminative touch [15, 16, 19–21], presumably like all mammals [22]. These receptors are in fact so ideally suited to convey socially relevant touch that they have become known as a "*social touch system*" [16, 23]. Since it is always reciprocal, social touch not only emotionally affects the receiver [14] but also the touch giver [24]. Touch is the primary modality for conveying intimate emotions [7, 14, 25]. To study the emotional impact of touch, validated and efficient affective self-report tools are needed.

In our digital age, human social interaction–including haptic interaction—becomes increasingly mediated. Most research on mediated haptic interaction has addressed the affective qualities of vibrotactile stimulation patterns [26–28]. However, recent technological developments like the embodiment of artificial entities [29], advanced haptic and tactile display technologies and standards ([30], including initial guidelines for mediated social touch [31]) also enable mediated social touch [32]. Mediated social touch may serve to enhance the affective quality of communication between geographically separated partners [33] and foster trust in and compliance with artificial entities [34]. Social touch between humans and touch-enabled agents and robots will become more significant with their increasing deployment in healthcare, teaching and telepresence applications [31, 35]. For all these applications, a complete understanding of the relation between the affective response space and the tactile stimuli is crucial for designing effective stimuli. Again, validated and efficient tools are needed to measure human affective response to tactile signals.

In accordance with the circumplex model of affect [36], the affective responses elicited by tactile stimuli vary mainly over the two principal affective dimensions of valence and arousal [28]. Most studies on the emotional response to touch apply two individual one-dimensional Likert scales [37, 38] or SAM (Self-assessment Mannikin [39]) scales [33, 34, 40] to measure both affective dimensions. Although the SAM is a validated and generally accepted tool, it has some practical drawbacks. The emotions it depicts are often misunderstood [41, 42]. Although the valence scale of the SAM is quite intuitive (the mannikin's facial expression varies from a frown to a smile), its arousal dimension (depicted as an 'explosion' in the mannikin's stomach) is frequently misinterpreted [43–45]. When used to measure the affective appraisal of touch, it appeared that the SAM's arousal scale is not evident for participants [40]. When participants do not fully understand the meaning of "*arousal*", they tend to copy their valence response to the arousal scale (an anchoring effect) [46]. To remedy this effect, participants are sometimes provided with additional explanations about this scale, emphasizing that arousal refers to emotional arousal [40]. The use of Likert scales involves cognitive effort, since they require the user to relate experienced emotions to numbers or labels on a scale. As a result, people with stroke related impairments [47] and children [48] are unable to correctly complete VAS scales whereas they can successfully use scales based on iconic facial expressions [49, 50]. The cognitive effort required for the use of VAS scales can also result in an emotion-cognition interaction, such that participants react faster and more consistently to stimuli that are rated as more unpleasant or more arousing than to stimuli that are rated as more pleasant and less arousing [37]. Also, SAM and Likert scales both require two successive assessments of valence and

arousal (along two individual scales), making their use more elaborate, error prone and time consuming. Previous studies therefore recognized the need to develop new rating scales for affective touch [51].

We recently presented a new intuitive and language-independent self-report instrument called the EmojiGrid (Fig 1): a square grid labeled with facial icons (emoji) expressing different levels of valence (e.g., angry face vs. smiling face) and arousal (e.g., sleepy face vs. excited face) [46]. In previous studies we found that participants can intuitively and reliably report their

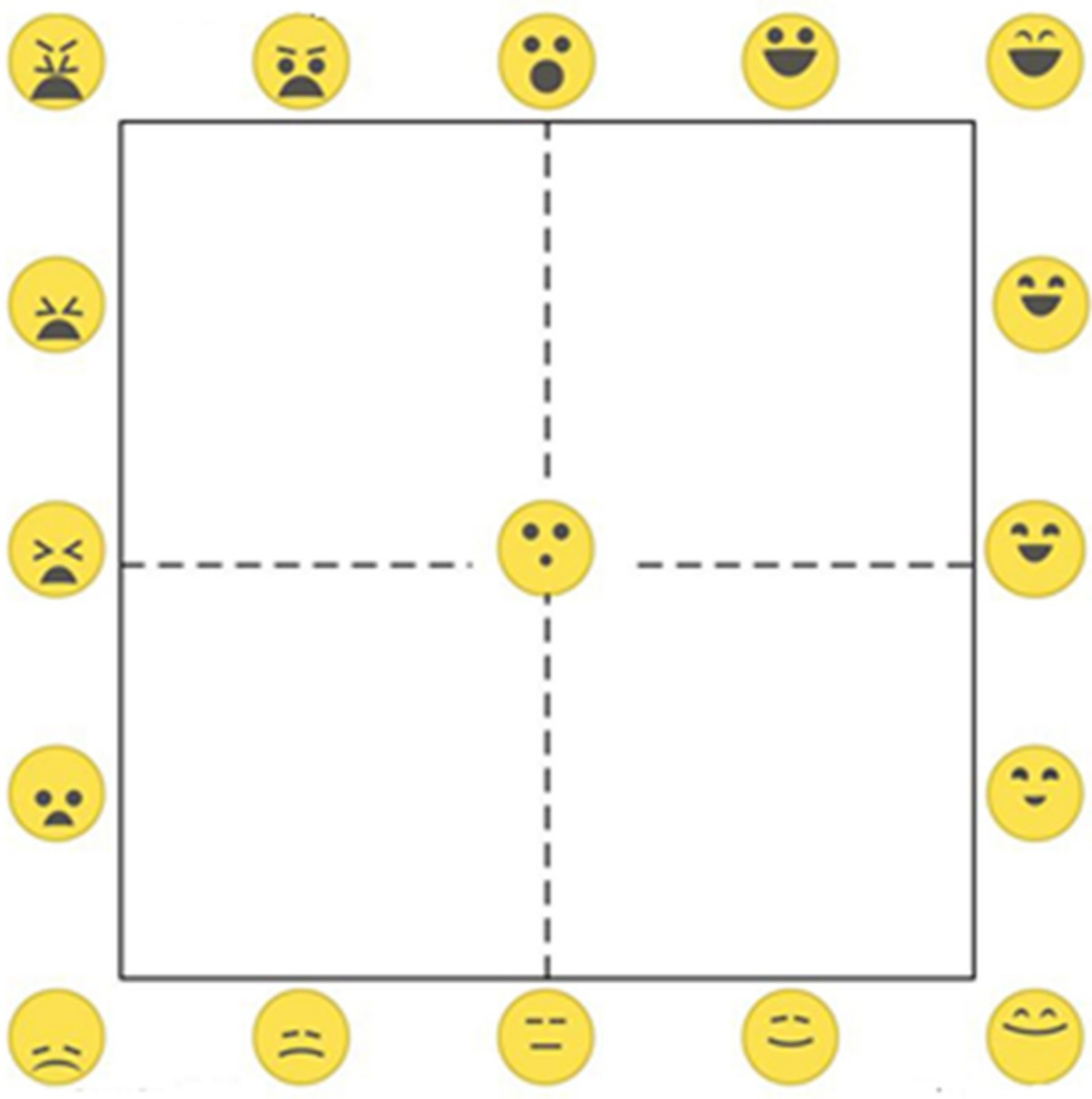

**Fig 1. The EmojiGrid.** The facial expressions of the emoji along the horizontal (valence) axis gradually change from unpleasant via neutral to pleasant, while the intensity of the facial expressions gradually increases in the vertical (arousal) direction.

emotional response with a single click on the EmojiGrid, even without any further instructions [46, 52–54]. This suggested that the EmojiGrid might also have more general validity as a self-report instrument to assess human affective responses. Given the intimate relationship between affective touch and affective facial expressions [55] we believe that the EmojiGrid may also be a useful tool to rate the affective appraisal of touch events.

In this study, we evaluated the convergent validity of the EmojiGrid as a self-report tool for the affective assessment of perceived touch events. We thereto measured perceived valence and arousal for various touch events in video clips from the validated Socio-Affective Touch Expression Database (SATED: [40]) using both the EmojiGrid and the validated SAM affective rating tool, and we compared the results. While social and non-social touch events elicit different neural activation patterns [56], it appears that the brain activity patterns elicited by imagined, perceived and experienced (affective) touch are highly similar [7, 19, 57–59]. The mere observation of social touch (relative to observation of object touch) activates a number of brain regions (including primary and secondary somatosensory cortices) in a somatotopical way [59], resulting in similar pleasantness ratings [19]. Mental tactile imagery recruits partially overlapping neural substrates in both the primary and secondary somatosensory areas [57]. The anterior insula, which has been implicated in anticipating touch and coding its affective quality [60], responds both to experienced and imagined touch [58]. To some extent, people experience the same touches as the ones they see (they have the ability to imagine how an observed touch would feel [61]): seeing other people's hands [62], legs [63], neck, or face [59] being touched activated brain regions that also respond when participants are touched on the same body part. Overall, these findings suggest that video clips showing touch actions are a valid means to study affective touch perception [64].

## 2 Methods and procedure

### 2.1 Stimuli

The stimuli used in this study are all 75 video clips from the validated Socio-Affective Touch Expression Database (SATED [40]). These clips represent 25 different dynamic touch events varying widely in valence and arousal. The interpersonal socio-affective touch events (N = 13) show people hugging, patting, punching, pushing, shaking hands or nudging each other's arm (Fig 2A). The object-based (non-social) touch events (N = 12) represent human-object interactions with motions that match those involved in the corresponding social touch events, and show people touching, grabbing, carrying, shaking or pushing objects like bottles, boxes, baskets, or doors (Fig 2B). Each touch movement is performed three times (i.e., by three different actors or actor pairs) and for about three seconds, resulting in a total of 75 video clips. All video clips had a resolution of 640×360 pixels.

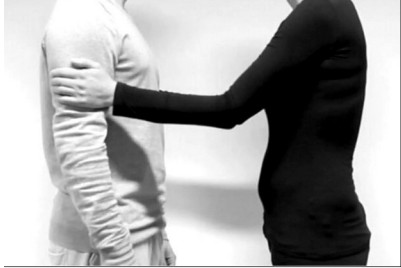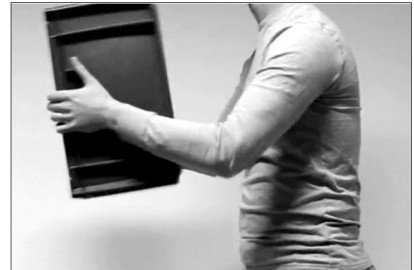

**Fig 2. Screenshots.** An interpersonal (socio-affective) touch event (left) and a corresponding object-based touch event (right).

## 2.2 Measures

**2.2.1 Demographics.** Participants reported their age, gender, and nationality.

**2.2.2 Valence and arousal.** In two independent online experiments, valence and arousal ratings were obtained for each of the 75 SATED video clips using either a 9-point SAM scale (Experiment 1) or the EmojiGrid (Experiment 2). The SAM is a pictorial self-report tool that enables users to rate the valence and arousal dimensions of their momentary feelings by selecting those humanoid figures that best reflect their own feelings. The EmojiGrid (Fig 1) is a square grid that is labeled with emoji showing different facial expressions. Each outer edge of the grid is labeled with five emoji, and there is one (neutral) emoji located in its center. The facial expressions of the emoji along a horizontal (valence) edge vary from disliking (unpleasant) via neutral to liking (pleasant), and their expression gradually increases in intensity along a vertical (arousal) edge. Users can report their affective state by placing a checkmark at the appropriate location on the grid. The EmojiGrid was first introduced in a study on food evoked emotions [46].

## 2.3 Participants

English speaking participants were recruited via the Prolific database (https://prolific.ac). A total of 65 participants (40 females, 25 males) aged between 18 and 35 (M = 27.5; SD = 5.1) participated in Experiment 1 (SAM rating). A total of 65 participants (43 females, 22 males) aged between 18 and 35 (M = 29.2; SD = 5.2) participated in Experiment 2 (EmojiGrid rating). The experimental protocol was reviewed and approved by the TNO Ethics Committee (Ethical Approval Ref: 2017–011) and was in accordance with the Helsinki Declaration of 1975, as revised in 2013 [65]. Participation was voluntary. After completing the study, all participants received a small financial compensation for their participation.

## 2.4 Procedure

The experiments were performed as (anonymous) online surveys created with the Gorilla experiment builder [66]. In each experiment, the participants viewed 75 brief video clips showing a different touch event and rated (using the SAM in Experiment 1 and the EmojiGrid in Experiment 2) for each video how the touch would feel. First, the participants signed an informed consent and reported their demographic variables. Next, they were introduced to either the SAM (Experiment 1) or the EmojiGrid (Experiment 2) response tool and were instructed how they could use this rating tool to rate their affective appraisal of each perceived touch event. For the SAM, we explained that the feelings expressed by the figures on the valence scale ranged from very unpleasant via neutral to very pleasant, while the figures on the arousal scale expressed feelings ranging from very relaxed/calm via neutral to very excited/stimulated. The instructions for the use of the SAM scale stated: "*Click on the response scales to indicate how pleasant (upper scale) and arousing (lower scale) the touch event feels*". No explanation was offered for the meaning of the facial icons along the EmojiGrid. The instructions for the use of the EmojiGrid stated: "*Click on a point inside the grid that best matches how you think the touch event feels*". No further explanation was given. In each experiment, the participants performed two practice trials to get familiar with the SAM or EmojiGrid and its use. Immediately after these practice trials the actual experiment started. The video clips were presented in random order. The rating task was self-paced without imposing a time-limit. After seeing each video clip, the participants responded by successively clicking on the valence and arousal scales of the SAM tool (Experiment 1; see Fig 3) or with a single click on the EmojiGrid (Experiment 2; see Fig 4). Immediately after responding the next video clip was presented. On average the experiment lasted about 10 minutes.

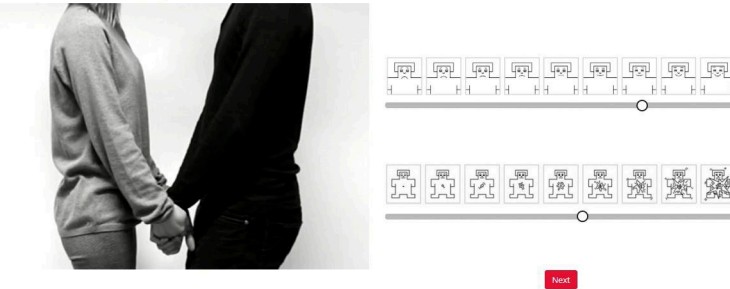

**Fig 3. Screen layout in Experiment 1.** Left: a screenshot of a movie clip showing an interpersonal (social) touch action. Right: The SAM valence (top) and arousal (bottom) rating scales.

**2.4.1 Data analysis.** IBM SPSS Statistics 26 (www.ibm.com) for Windows was used to perform all statistical analyses. Intraclass correlation coefficient (ICC) estimates and their 95% confident intervals were based on a mean-rating (k = 3), absolute agreement, 2-way mixed-effects model [67, 68]. ICC values less than .5 are indicative of poor reliability, values between .5 and .75 indicate moderate reliability, values between .75 and .9 indicate good reliability, while values greater than .9 indicate excellent reliability [67]. For all other analyses, a probability level of p < .05 was considered to be statistically significant.

For each of the 25 different touch scenarios, we computed the mean valence and arousal responses over all three of its representations (three actor pairs) and over all participants. We used Matlab 2019b (www.mathworks.com) to investigate the relation between the (mean) valence and arousal ratings and to plot the data. The Curve Fitting Toolbox (version 3.5.7) in Matlab was used to compute a least-squares fit of a quadratic function to the data points.

## 3 Results

Table 1 lists the median valence and arousal ratings obtained with the SAM and the Emoji-Grid, for all touch events and for social and non-social events separately. First, we verified the results reported by Masson & Op de Beeck [40] for their SATED database.

A Mann-Whitney U test revealed that mean SAM valence ratings were indeed significantly higher for social touch scenarios classified in SATED as positive or pleasant (Mdn = 6.96, MAD = .33., n = 6) than for those that were classified as negative or unpleasant (Mdn = 3.11, MAD = .26, n = 6), U = 0, z = -2.89, p = .004. Also, mean SAM valence ratings for the positive social touch scenarios were indeed significantly higher than the corresponding ratings for

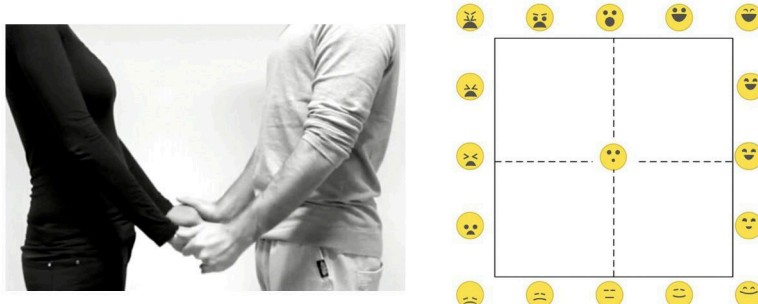

**Fig 4. Screen layout in Experiment 2.** Left: a screenshot of a movie clip showing an interpersonal (social) touch action. Right: The EmojiGrid rating tool.

**Table 1. Median (MAD) valence and arousal ratings, obtained with the SAM and the EmojiGrid, for the (positive, negative and all) social and object touch scenarios from the SATED database.**

| Scenarios | | SAM | | EmojiGrid | |
|---|---|---|---|---|---|
| | | Valence | Arousal | Valence | Arousal |
| Social touch | positive | 6.96 (.29) | 4.55 (.25) | 7.60 (.24) | 5.08 (.76) |
| | negative | 3.11 (.26) | 4.94 (.21) | 2.64 (.36) | 5.90 (0.53) |
| | overall | 5.55 (1.87) | 4.80 (.30) | 6.05 (2.25) | 5.52 (.79) |
| Object touch | positive | 5.12 (.20) | 3.76 (.13) | 5.65 (.22) | 3.67 (.12) |
| | negative | 4.04 (.40) | 4.26 (.18) | 4.14 (.58) | 4.31 (.21) |
| | overall | 4.87 (.45) | 3.94 (.23) | 5.45 (.62) | 4.06 (.37) |

object-based touch scenarios (Mdn = 5.12, MAD = .20, n = 6, U = 0, z = -2.89, p = .004), while mean SAM valence ratings for the negative social touch scenarios were significantly lower than the corresponding ratings for the object-based touch scenarios (Mdn = 4.04, MAD = .40, n = 6, U = 0, z = -2.88, p = .004).

Then, we compared the mean SAM arousal ratings between social touch and object touch. The results revealed that social touch was indeed overall rated as more arousing (Mdn = 4.80, MAD = .35) than object-based (non-social) touch (Mdn = 3.94, MAD = .18, U = 7.0, z = -3.75, p < = 0.000), as reported by Masson & Op de Beeck [40]. Also, mean SAM arousal ratings for the positive social touch scenarios (Mdn = 4.55, MAD = .25) were significantly higher than the corresponding ratings for the object-based touch scenarios (Mdn = 3.76, MAD = .13, n = 6, U = 0, z = -2.88, p = .004), while mean SAM arousal ratings for the negative social touch scenarios (Mdn = 4.94, MAD = .21) were significantly higher than the corresponding ratings for the object-based touch scenarios (Mdn = 4.26, MAD = .18, n = 6, U = 1, z = -2.72, p = .006). Thus, it appears that social touch is more arousing than object-based (non-social) touch, suggesting that interpersonal touch is experienced as being more intense. Summarizing, our current findings fully agree with and confirm the corresponding results reported previously by Masson & Op de Beeck [40].

To quantify the agreement between the SAM ratings obtained in this study and those reported by Masson & Op de Beeck [40], we computed Intraclass Correlation Coefficient (ICC) estimates with their 95% confidence intervals for the mean valence and arousal ratings between both studies. For valence we find an ICC value of .98 [.96 -.99], indicating excellent reliability. For arousal, the ICC value is .59 [.07 - .82], indicating moderate reliability.

Next, we continued to investigate the agreement between the ratings obtained in this study with the SAM and with the EmojiGrid self-report tools.

Fig 5 shows the correlation plots between the mean subjective valence and arousal ratings obtained with the EmojiGrid and with the SAM. This figure shows that the mean valence ratings for all touch events closely agree between both studies, while the original classification of the social touch scenarios by Masson & Op de Beeck [40] into positive (scenarios 1–6), negative (scenarios 8–13) and neutral (scenario 7) scenarios also holds in this result. For social touch, the EmojiGrid appears more sensitive to arousal than the SAM: over the same valence range, the variation in mean arousal ratings is larger for the EmojiGrid than for the SAM.

Fig 6 shows the relation between the mean valence and arousal ratings for all 25 different SATED scenarios, as measured with the 9-point SAM scale and with the EmojiGrid. The curves represent least-squares quadratic fits to the data points. The adjusted R-squared values are respectively .74 and .88, indicating good fits. This figure shows that the relation between the mean valence and arousal ratings obtained with both self-assessment methods (SAM and EmojiGrid) is closely described by a quadratic (U-shaped) relation at the nomothetic (group)

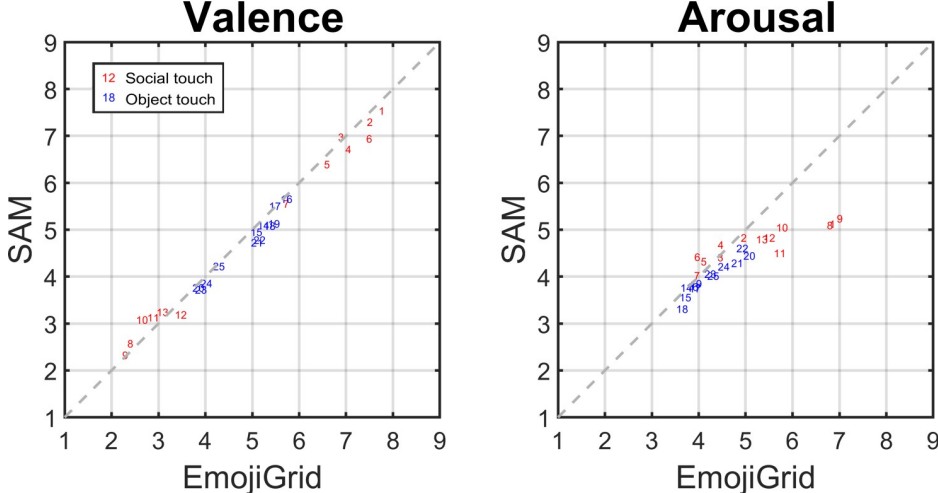

**Fig 5. Correlation plots.** The relationship between the valence (left) and arousal (right) ratings provided with a 9-point SAM scale and with the EmojiGrid. The numbers correspond to the original scenario identifiers in the SATED database [40]. The social touch scenarios (red labels) 1–6, 7 and 8–13 were originally classified as positive, neutral and negative, and the object touch scenarios (blue labels) 14–25 as neutral.

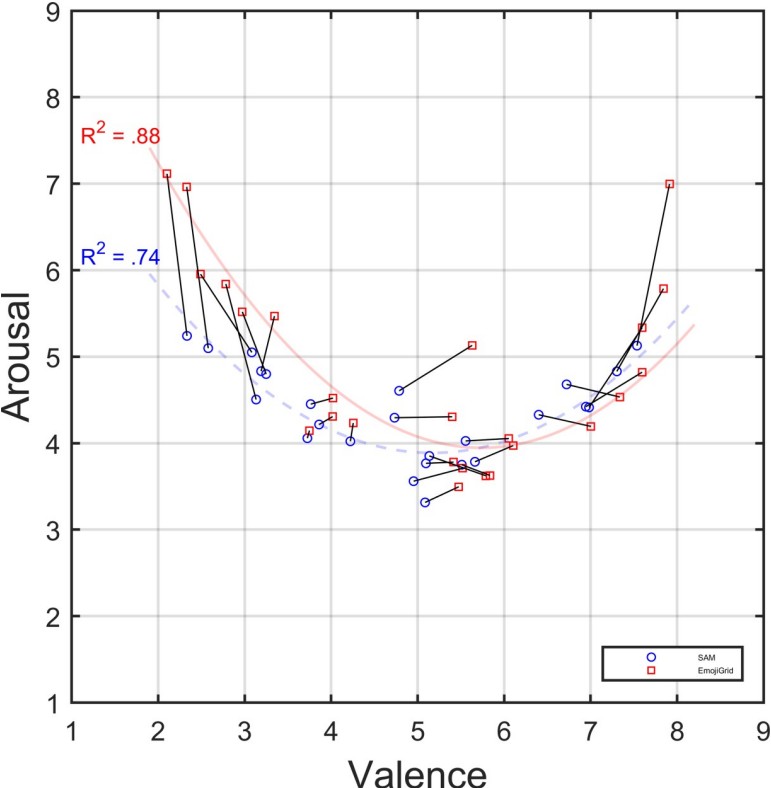

**Fig 6. Relation between the mean valence and arousal ratings.** Mean valence and arousal ratings for affective touch video clips from the SATED database, obtained with the a 9-point SAM rating scale (blue dots) and with the EmojiGrid (red dots). The curves represent fitted polynomial curves of degree 2 using a least-squares regression between valence and arousal. The line segments connect ratings for the same video clips.

**Table 2. Intraclass correlation coefficients with their 95% confidence intervals for mean valence and arousal ratings obtained with the SAM and with the EmojiGrid (this study), for video clips from the SATED database [40].**

| Stimuli | N | ICC Valence | ICC Arousal |
|---|---|---|---|
| All touch events | 25 | .99 [.97–1.0] | .78 [.35 – .91] |
| Social touch events | 13 | 1.0 [.99 – 1.0] | .61 [-.13 – .87] |
| Object touch events | 12 | .97 [.44–1.0] | .88 [-.04 – .97] |

level: touch events scoring near neutral on mean valence have the lowest mean arousal ratings, while touch events scoring either high (pleasant) or low (unpleasant) on mean valence show higher mean arousal ratings. For increasing absolute valence (absolute difference from neutral valence) the EmojiGrid appears increasingly more sensitive to variations in arousal (the length of the line segments in Fig 6 systematically increases for increasing absolute valence).

To quantify the agreement between the ratings obtained with the 9-point SAM scales and with the EmojiGrid we computed Intraclass Correlation Coefficient (ICC) estimates with their 95% confidence intervals for the mean valence and arousal ratings obtained with both tools, for all touch events and for social and non-social events separately (see Table 2). The valence ratings show excellent reliability in all cases. The arousal ratings show good reliability for all touch events and for object-based touch events, and moderate reliability for social touch events.

## 4 Conclusion and discussion

The valence ratings obtained with the EmojiGrid show excellent agreement with valance ratings obtained with a 9-point SAM scale, both for inter-human (social) and object-based touch events. The arousal ratings obtained with the EmojiGrid show good agreement with arousal ratings obtained with a 9-point SAM scale, both overall and for object-based touch events. However, for social touch events both arousal ratings only show moderate agreement. This results from the fact that the sensitivity of the EmojiGrid appears to increase with increasing absolute valence (deviation from neutral). It appears that the ratings obtained with the SAM tool show a ceiling effect, possibly because participants hesitate to use the extreme 'explosion' icons near the higher end of the SAM arousal scale while they are still confident to use the more extreme facial expressions of the emoji on the arousal axis of the EmojiGrid.

Our results also replicate the U-shaped (quadratic) relation between the mean valence and arousal ratings, as reported in the literature [40].

Summarizing, we conclude that the EmojiGrid appears to be a valid graphical self-report instrument for the affective appraisal of perceived social touch events, which appears to be more sensitive to variations in arousal than the SAM at more extreme levels of valence. A previous study found that slow brush stroking on the forearm and the palm elicited significantly different brain activations, while individual VAS pleasantness and intensity scales were not sufficiently sensitive to distinguish between these two types of touch [69]. Since the EmojiGrid combines both valence and arousal, it would be interesting to investigate whether this instrument is sensitive enough to differentiate between these kinds of touch. However, more studies using different emotion elicitation protocols are required to fully assess the validity of this new tool.

A limitation of this study is that we did not investigate the affective appraisal of real touch events. However, given the ability of people to imagine how an observed touch would feel [61], we expect that the EmojiGrid will also be useful as a self-report tool for the affective appraisal of experienced touch events. Further research is needed to test this hypothesis.

## Supporting information

**S1 File. Mean and SD values measured with SAM and EmojiGrid for each of the 25 different dynamic touch events.**
(XLSX)

## Author Contributions

**Conceptualization:** Alexander Toet.

**Data curation:** Alexander Toet.

**Formal analysis:** Alexander Toet, Jan B. F. van Erp.

**Investigation:** Alexander Toet, Jan B. F. van Erp.

**Methodology:** Alexander Toet, Jan B. F. van Erp.

**Resources:** Alexander Toet, Jan B. F. van Erp.

**Software:** Alexander Toet.

**Supervision:** Alexander Toet, Jan B. F. van Erp.

**Validation:** Alexander Toet.

**Visualization:** Alexander Toet.

**Writing – original draft:** Alexander Toet, Jan B. F. van Erp.

**Writing – review & editing:** Alexander Toet, Jan B. F. van Erp.

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
