## [Decision Letter · Decision Letter 0]

8 Jul 2020

PONE-D-20-09067

The EmojiGrid as a rating tool for the affective appraisal of touch

PLOS ONE

Dear Dr. Toet,

Thank you for submitting your manuscript to PLOS ONE. After careful consideration, we feel that it has merit but does not fully meet PLOS ONE’s publication criteria as it currently stands. Therefore, we invite you to submit a revised version of the manuscript that addresses the points raised during the review process.

We look forward to receiving your revised manuscript.

Kind regards,

Enzo Pasquale Scilingo, Ph.D.

Academic Editor

PLOS ONE

Journal Requirements:

Additional Editor Comments (if provided):

The manuscript reports on a new self-report tool for rating affective dimension of emotional touch. It's of interest and technically sound, but it needs some revisions before it might be considered for publication. Please follow carefully the recommendations from the reviewers accordingly.

Reviewers' comments:

Reviewer's Responses to Questions

**Comments to the Author**

1. Is the manuscript technically sound, and do the data support the conclusions?

Reviewer #1: Yes

Reviewer #2: Yes

2. Has the statistical analysis been performed appropriately and rigorously? 

Reviewer #1: Yes

Reviewer #2: I Don't Know

3. Have the authors made all data underlying the findings in their manuscript fully available?

Reviewer #1: Yes

Reviewer #2: Yes

4. Is the manuscript presented in an intelligible fashion and written in standard English?

Reviewer #1: Yes

Reviewer #2: Yes

5. Review Comments to the Author

Reviewer #1: The paper describes the validation of a self-report tool, which captures both valence and arousal ratings in a single response, for the appraisal of social and object-based touch. Across two online studies the authors show that ratings on their EmojiGrid show a high level of agreement with ratings of the same stimuli (social and object touch videos) on the Self-Assessment Manikin – an established scale which requires two separate responses to gather valence and arousal data. Overall, the paper is clearly presented, and the conclusions reached supported by the data presented. I have only a few minor comments that I feel could improve the clarity of the paper.

Abstract (lines 19-21) “We conclude that the EmojiGrid can also serve as a valid and efficient graphical self-report instrument to measure human affective response to a wide range of (possibly mediated) tactile signals.” In this context, without reference to any relevant background literature, it isn’t clear what “mediated” tactile signals are. The authors could consider removing the term from the abstract or revising to make its meaning clearer. When the same term is used in the introduction it is a little more clear because of the context the term is being used in.

Introduction (lines 24 – 40)- In this first paragraph of the introduction the authors should more explicitly define what they mean by affective touch. Initially it seems to refer to an emotional evaluation of a tactile stimulus – eg the properties of a fabric. However, it is then related to the specific response properties of a class of cutaneous afferent found in the hairy skin of mammals which responds optimally to low force and velocity touch. Affective touch does not necessarily have to mean C-tactile afferent activating touch and the authors are best to define it with relevance to their own study. For example, exploration and evaluation of a soft material with the fingertips and palms of the hands might be termed affective touch but it doesn’t involve activation of C-tactile afferents.

Relatedly on line 87 – Just because touch is affective it doesn’t have to be social. Is this line referring to discriminative and affective touch, or social and non-social touch, having different neural activations?

Methods – Data Analysis section (lines 187-196): The authors use non-parametric analysis of their ratings data. The approach used to determine assumptions for parametric testing were not met should be reported.

Discussion – The discussion is very short and there is scope to go into more depth on the potential utility of this scale for future tactile research.

For example, given it combines both valence and arousal, could it be a more sensitive measure for differentiating between evaluations of touch than standard visual analogue scales? This paper provides an example of limitations of a VAS in differentiating between touch on different skin sites - McGlone, F., Olausson, H., Boyle, J. A., Jones‐Gotman, M., Dancer, C., Guest, S., & Essick, G. (2012). Touching and feeling: differences in pleasant touch processing between glabrous and hairy skin in humans. European Journal of Neuroscience, 35(11), 1782-1788.

Minor comments

Line 27 -28 – “This affective component of touch plays a significant role in social communication.” – please provide a reference to support this statement.

Line 33 – suggest “persons” is changed to “people”

Line 83-84 – please provide a reference for the “validated affective image database” referred to here.

Line 86-87 – please give term in full on first use before using acronym SATED and provide a reference.

Line 221 – p is reported as equal to 0?

Reviewer #2: The manuscript investigates the validity of a new self-report tool for rating affective dimension of emotional touch. The paper is clear but there are minor mistakes in the English that needs to be corected.

Although, the method seems technically sound,, I have some concerns in other sections which I list below.

1) Introduction:

1.1 The introduction addresses general points about affective touch. However, it lacks more detailed literature review on the state of the art studies investigating affective dimension of touch.

As an example, the authors introduce in lines 57-70, the SAM test and Likert scales as the state-of-the-art self-assessment tools used in affective studies. I suggest to enlarge this paragraph by discussing the studies using these tools in rating affective touch, the main advantages and disadvantages of these methods and how previous studies tackled the limitations in rating arousal level.

1.2.

The authors justify the use of video clips in eliciting the sense of affective touch to the subjects by evidences in Lines 81 to 94 mentioning " it appears that the brain activity patterns elicited by imagined, perceived and experienced (affective) touch are highly similar [11, 12, 45-47].". I suggest the authors to expand this sentence by discussing more details on the conclusions of these studies. It is not clear and there is no detailed discussion on how the choice of eliciting affective touch by perceiving it could have the similar effect as a real elicitation and what limitations it has.

2) Results

This section lacks readability, a table reporting the statistical analysis (Median and MAD values and p-values) for a comparative visualisation of the results is missing.

The general outcome of this study suggesting the improvement in rating arousal and valence scales with the proposed self-report tool rather than the standard SAM test is promising for all affective future studies. However, much more investigations in different elicitation protocols should be done to validate the new tool.

6. PLOS authors have the option to publish the peer review history of their article (what does this mean?). If published, this will include your full peer review and any attached files.

Reviewer #1: No

Reviewer #2: **Yes: **Shadi Ghiasi

---

## [Author Response · Author response to Decision Letter 0]

13 Jul 2020

See attached Reply-to-reviewers

---

## [Decision Letter · Decision Letter 1]

5 Aug 2020

The EmojiGrid as a rating tool for the affective appraisal of touch

PONE-D-20-09067R1

Dear Dr. Toet,

We’re pleased to inform you that your manuscript has been judged scientifically suitable for publication and will be formally accepted for publication once it meets all outstanding technical requirements.

Kind regards,

Enzo Pasquale Scilingo, Ph.D.

Academic Editor

PLOS ONE

Additional Editor Comments (optional):

Reviewers' comments:

Reviewer's Responses to Questions

**Comments to the Author**

1. If the authors have adequately addressed your comments raised in a previous round of review and you feel that this manuscript is now acceptable for publication, you may indicate that here to bypass the “Comments to the Author” section, enter your conflict of interest statement in the “Confidential to Editor” section, and submit your "Accept" recommendation.

Reviewer #1: All comments have been addressed

Reviewer #2: All comments have been addressed

2. Is the manuscript technically sound, and do the data support the conclusions?

Reviewer #1: Yes

Reviewer #2: Yes

3. Has the statistical analysis been performed appropriately and rigorously? 

Reviewer #1: Yes

Reviewer #2: I Don't Know

4. Have the authors made all data underlying the findings in their manuscript fully available?

Reviewer #1: Yes

Reviewer #2: Yes

5. Is the manuscript presented in an intelligible fashion and written in standard English?

Reviewer #1: Yes

Reviewer #2: Yes

6. Review Comments to the Author

Reviewer #1: The authors have addressed all of my questions.

However, my comment about non-parametric analysis referred not to the ICC analysis but the Mann Whitney U tests, which I assume they used because the ratings data are ordinal. I wasn't questioning the validity of the choice, I merely suggested adding a statement in the data analysis section to that effect.

Reviewer #2: The authors have addressed all the raised comments and the paper is more clear now. I suggest to accept without further revision.

7. PLOS authors have the option to publish the peer review history of their article (what does this mean?). If published, this will include your full peer review and any attached files.

Reviewer #1: No

Reviewer #2: **Yes: **Shadi Ghiasi

---

## [Editor Report · Acceptance letter]

6 Aug 2020

PONE-D-20-09067R1 

The EmojiGrid as a rating tool for the affective appraisal of touch 

Dear Dr. Toet:

I'm pleased to inform you that your manuscript has been deemed suitable for publication in PLOS ONE. Congratulations! Your manuscript is now with our production department. 

Kind regards, 

on behalf of

Professor Enzo Pasquale Scilingo 

Academic Editor

PLOS ONE